# The Heart of the Killer Whale: Description of a Plastinated Specimen and Review of the Available Literature

**DOI:** 10.3390/ani12030347

**Published:** 2022-01-31

**Authors:** Rafael Latorre, Jean-Marie Graïc, Stephen A. Raverty, Federico Soria, Bruno Cozzi, Octavio López-Albors

**Affiliations:** 1Department of Anatomy and Comparative Pathology, University of Murcia, 30100 Murcia, Spain; latorre@um.es (R.L.); albors@um.es (O.L.-A.); 2Department of Comparative Biomedicine and Food Science, University of Padova, 35020 Padova, Italy; bruno.cozzi@unipd.it; 3Animal Health Center, British Columbia Ministry of Agriculture, Abbotsford, BC V3G 2M3, Canada; stephen.raverty@gov.bc.ca; 4Department of Endoscopy-Endourology, Jesús Usón Minimally Invasive Surgery Centre, 10071 Cáceres, Spain; fsoria@ccmijesususon.com

**Keywords:** killer whale, heart, papillary muscles, coronary arteries, ventricles, atria

## Abstract

**Simple Summary:**

The killer whale (*Orcinus orca*, Linnaeus, 1958) is an odontocete and is the largest member of the family *Delphinidae*. Free-ranging animals are capable of considerable physical efforts, either as acute bursts or sustained speeds during foraging, diving or protracted long-distance migrations. In this article, the morphology of a plastinated heart of a killer whale and functional adaptations of the gross anatomy in the context of a variety of physiologic demands is evaluated. The four chambers, their content, respective openings and communicating passages are defined based on a specimen used for plastination and thus available for extended and detailed anatomic studies.

**Abstract:**

The killer whale (*Orcinus orca*, Linnaeus, 1958) is the largest extant delphinid. Despite its worldwide distribution in the wild and in dolphinariums, its anatomy remains relatively poorly described. In the present study, we describe the detailed morphology of a plastinated killer whale heart. The gross description of the arteries and veins reaching the organ and its coronary vessels are reported. Additional endoscopy and CT (computed tomography) scanning were performed to provide extensive measurements of its parts. In many aspects, the killer whale heart conformed to other delphinid heart descriptions, including position, relative size and shape and specific features such as extensive papillary muscles, trabecular endocardium and *trabecula septomarginalis*. These characteristics are representative of the delphinid family, suggesting that its functions and capacities are similar to that of other, smaller, dolphins and help understand the conditions in which these predators exert their remarkable physical performance necessary for their survival.

## 1. Introduction

The killer whale (*Orcinus orca*, Linnaeus 1758) is the largest member of the family Delphinidae. Highly dimorphic in size, the body length varies from 5 m in females to over 9 m in males, and the body weight from 3 tons in females to over 8 tons in males. Several subpopulations of killer whales (“ecotypes”) are identified based on geographical location and prey preference. All killer whales, however, are capable of astounding physical performances, necessary to hunt in groups, such as predation by transient killer whales of larger Mysticetes (baleen whales), or with bursts of great speed observed in killer whales that hunt for pinnipeds, salmon or tuna. Although all ecotypes of orcas do not need to dive deep for their survival, some [1] are capable of extended breath-holding dives reaching considerable depths (see Table 1).

**Table 1 animals-12-00347-t001:** Key biological data on cardio-respiratory functions in killer whales.

Data	Value	Condition	Reference
Breaths/min	0.9 (0.8–1.1)	at rest	[2]
	1.3 ± 0.6		[3]
Heart rate	44 ± 2.1 bpm (22–75)	at rest	[4]
	58 ± 1.2 bpm (23–81)	underwater swim	
Apnea	up to 9.6–13.3′	voluntary, shallow water	[5]
	13.57′–15.9′	ocean dive	[6]
Max dive depth	767.5 m	ocean dive	[6]

Historically, the anatomy and physiology of killer whales has received relatively little attention [7,8].

The anatomy of the killer whales has not yet been fully characterized and is typically based on anatomic features associated with opportunistic post-mortem examinations or extrapolation from smaller, more common odontocetes, such as dolphins and porpoises [9,10,11]. In this article, we characterize the gross anatomy and detail adaptations in the killer whale heart. A comprehensive description of the topography and anatomy of the heart of dolphins has been reported for the bottlenose dolphin (*Tursiops truncatus*, see [12]) and other species (see [13]). The most salient anatomic differences identified between terrestrial mammals and representatives of the family *Delphinidae* were: (1) a broader thorax, (2) a ventrodorsally flattened heart, (3) flush with the sternum and (4) slightly more cranial within the thoracic cavity. As a result, the lungs in marine mammals are positioned more dorsal within the thoracic cavity [12,14,15]. Deep coronary and atrioventricular grooves demarcate the four chambers. Descriptions of the *Foramen ovale* and aortic arch of *O. orca* have previously been documented by McDonald et al. [9] and Slijper [14], respectively.

The total mass of circulating blood of members of the family *Delphinidae* is approx. 7.4% of total body mass (thus between 20 to 60 L in the killer whale) [16,17]. The weight of a normal heart in the bottlenose dolphin is 0.93% of the total body weight of the bottlenose dolphin and approx. 0.5 to 1.3% of the total body mass of other adult members of the family (thus between 15 and 40 kg in the killer whale) [12,14,16,18]. Both the heart mass and blood volume have been reported to be lower than those of terrestrial mammals [19]. However, more recent work seems to show a certain variability [7,20]; for a review on the mammalian heart, see also [20].

The recent plastination and museum accessioning of an adult male killer whale heart [21] afforded a unique opportunity to better define the gross and CT (Computed Tomography) cardiac anatomy of a large delphinid and contrast these features with other cetacean and terrestrial species.

## 2. Materials and Methods

The heart analyzed in this study was recovered from an adult male southern resident killer whale (SRKW), identified as L95. The SRKW ecotype are designated as endangered under the Committee on the Status of Endangered Wildlife in Canada (COSEWIC, 2001 and 2008) and United States Endangered Species Act (ESA, 2006).

The animal was observed dead and floating offshore on 30 March 2016, by the Department of Fisheries and Oceans (DFO) Cetacean Research program at 49°32.5′ N, 127°13.8′ W, west of Nootka Sound, off the west coast of Vancouver Island, British Columbia [21], and subsequently towed to a secluded and secure area. A necropsy was performed by standardized killer whale necropsy protocol [22]. The animal presented in advanced post-mortem decomposition (code 3.5–4.0) and was in fair to moderate nutritional condition. Firstly, the skin and blubber were removed from the flank. Then, an incision in the abdominal musculature was extended along the caudal limit of the costochondral arch, to allow the examination of the internal abdominal viscera. The diaphragm was subsequently pierced to assess negative pressure in the pleural space and the ribs were disarticulated from the costochondral and costovertebral joints and removed. Topography and gross pathology of the lungs, heart and mediastinum were initially assessed in situ. The pericardium was subsequently incised to expose the heart and the greater vessels were identified and then sectioned: on the right aspect of the heart, both the vena cava and right pulmonary arteries and veins; and on the left side, the ascending aorta and left pulmonary arteries and veins. The heart mass was 24 kg.

Representative tissue samples were collected at the time of necropsy and histopathology and ancillary diagnostic studies determined that the cause of death was due to a generalized fungal infection (mucormycosis) emanating from a previously deployed satellite tag with retained petals at the base of the dorsal fin [23].

The heart was frozen, transported to the Royal Ontario Museum (ROM) and maintained at −80 °C for later processing for plastination [21].

Fixation of the heart was performed at the Royal Ontario Museum (Toronto, ON, Canada) while slow thawing according to previously published protocols for whale hearts [24,25,26,27]. Details of the particular fixation and plastination protocol of this specimen have been published in a previous paper [21]. After fixation, the heart was securely packaged en masse and shipped to the Plastination Lab of the University of Murcia, Spain with appropriate CITES permits and documentation for further processing. Before dehydration, the coronary arteries were injected with red silicon (RTV Dow Corning^®^ Midland, MI, USA; color Biodur^®^ AC52, Biodur Products GmBH, Heidelberg, Germany).

A week of room temperature precuring facilitated partial dissection of the *lamina visceralis* (epicardium) of the *pericardium serosum* in the coronary sulcus and paraconal interventricular sulcus. The myocardium was incised, and a window was opened through the wall of the right ventricle to expose the internal cardiac morphology. Final curing with hardener component S6 (Biodur^®^, Biodur Products GmBH, Heidelberg, Germany) was performed in a curing chamber for two more weeks. The total curing time was 3 weeks.

Due to the quality and unique value of this specimen, imaging studies with endoscopic and radiologic approaches were performed in the Minimally Invasive Surgery Center Jesús Usón (Cáceres, Spain) within a week after plastination was completed. A flexible endoscope (Fujinon 200, EPX-2200 processor, endoscope EG250WR5) was used to explore the lumen of the heart cavities. The endoscopic study started through the right atrium to visualize the endoluminal aspect of the right auricle, right ventricle and right atrioventricular valve as well as the pulmonary valve and pulmonary trunk. After the examination of the right side of the heart, the endoscopic exploration continued through the left atrium, and the endoluminal aspect of the left auricle and the left ventricle was assessed. The left atrioventricular valve and aortic valve were also visualized. The radiographic study of the heart was performed by computerized tomography (CT) scan (Philips BRILLIANCE CT-6). The scan parameters were 0.6 mm minimum slice thickness (0.6 mm) and full-body speed (360° in 0.75 s). The dorsoventral axis (transversal plane) was used to obtain the original sequential imaging. The images from the latero-lateral axis (sagittal plane) and the craniocaudal axis (horizontal plane) were reconstructed from the transversal plane images, such that the three spatial plane images were available for the description of the internal anatomy and overall morphology of the specimen.

The CT scan images were used to document the morphometry of relevant anatomical landmarks and 3D rendering with the OsiriX Lite^®^ software (v.12.0.1, Pixmeo, Geneva, Switzerland). Besides individual measurements of specific structures, measurements of large anatomical areas, such as the length and thickness of the interventricular septum or the parietal wall and lumen of ventricles, were determined by specific CT images. These reference planes included a CT sagittal plane incorporating the two ventricles and aortic sinus, and CT cross-sectional images at three different levels of the heart: Level I: a transversal section just distal to the atrioventricular valves; Level II: a transversal Section 10 cm distal from level I; and Level III: a transversal section 10 cm distal from level II (Figure 4).

## 3. Results

The plastination method (cold S10 technique) [28,29] allowed the preservation of the heart of the L95 SRKW (*O. orca*). Since the initial plastination in 2016, the general morphology of the cavities and the original topography of the main vessels, including the coronary arteries, are still evident (Figure 1, Figure 2 and Figure 3).

**Figure 1 animals-12-00347-f001:**
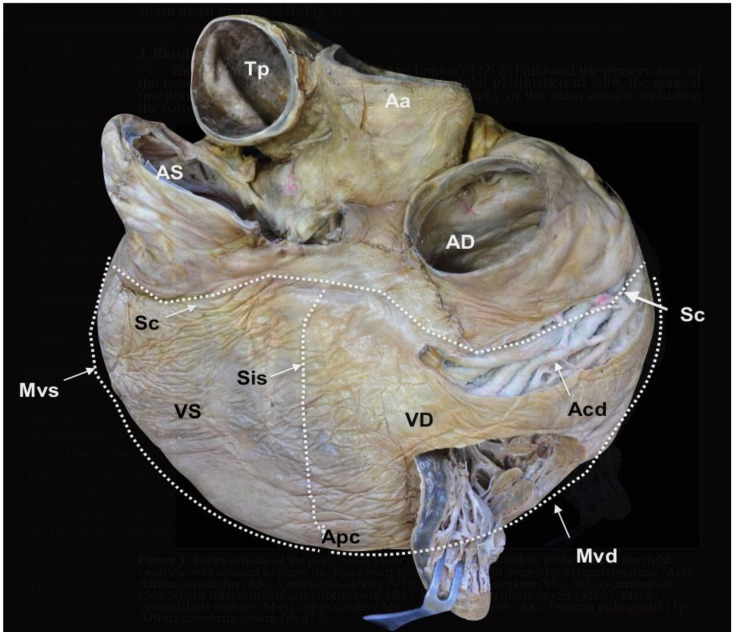
*Facies atrialis* of the plastinated killer whale heart. The parietal wall of the *Ventriculus dexter* (VD) is partially open. *Atrium dextrum* (AD), *Atrium sinistrum* (AS), *Ventriculus sinister* (VS), *Sulcus coronarius* (Sc), *Sulcus interventricularis subsinuosus* (Sis), *Margo ventricularis dexter* (Mvd), *Margo ventricularis sinister* (Mvs), *Apex cordis* (Apc), *Aorta ascendens* (Aa), *Truncus pulmonalis* (Tp), *Arteria coronaria dextra* (Acd) (modified from [21] with permission).

**Figure 2 animals-12-00347-f002:**
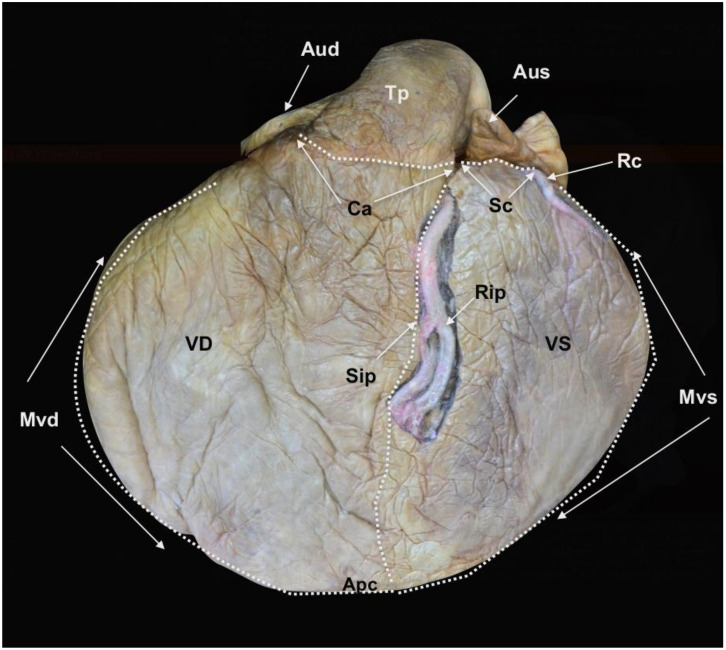
*Facies auricularis* of the plastinated killer whale heart. *Auricula dextra* (Aud), *Auricula sinistra* (Aus), *Ventriculus dexter* (VD), *Ventriculus sinister* (VS), *Sulcus coronarius* (Sc), *Conus arteriosus* (Ca), *Sulcus interventricularis paraconalis* (Sip), *Margo ventricularis dexter* (Mvd), *Margo ventricularis sinister* (Mvs), *Apex cordis* (Apc), *Truncus pulmonalis* (Tp), *Arteria coronaria sinistra*: *Ramus interventricularis paraconalis* (Rip), *Ramus circumflexus* (Rc) (modified from [21] with permission).

**Figure 3 animals-12-00347-f003:**
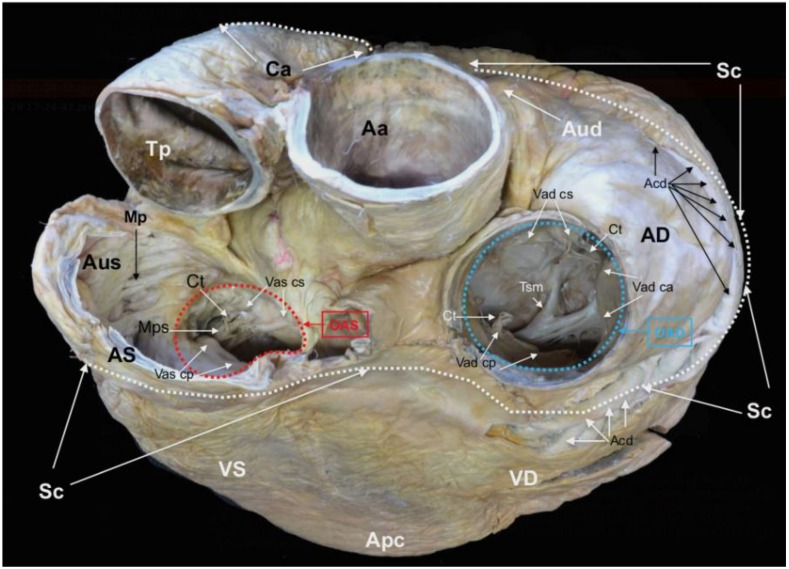
*Basis cordis* of the plastinated killer whale heart. *Aorta ascendens* (Aa), *Truncus pulmonalis* (Tp), *Auricula dextra* (Aud), *Auricula sinistra* (Aus), *Sulcus coronarius* (Sc), *Conus arteriosus* (Ca), *Ventriculus dexter* (VD), *Ventriculus sinister* (VS), *Apex cordis* (Apc). *Atrium dextrum* (AD), *Ostium atrioventriculare dextrum* (OAD), *Valva atrioventricularis dextra* (Vad) (*Valva tricuspidalis*): *Cuspis angularis* (Vad ca), *Cuspis parietalis* (Vad cp), *Cuspis septalis* (Vad cs), *Trabecula septomarginalis* (Tsm), *Atrium sinistrum* (AS), *Musculi pectinati* (Mp), *Ostium atrioventriculare sinistrum* (OAS), *Valva atrioventricularis sinistra* (Vas) (*Valva mitralis*): *Cuspis parietalis* (Vas cp), *Cuspis septalis* (Vas cs), *Musculus papillaris subauricularis* (Mps), *Chordae tendineae* (Ct), *Arteria coronaria dextra* (Acd) (modified from [21] with permission).

The external and internal morphology of the heart was assessed in detail by direct observation of the specimen (Figure 1, Figure 2 and Figure 3), combined with CT imaging (Figure 4, Figure 5 and Figure 6) and endoscopy (Figure 7, Figure 8 and Figure 9). The radiopacity of the plastinated specimen not only depicted intracavital structures but also intramural details of the myocardium. The description of the coronary vessels was also enhanced by the presence of RTV silicone injected in the coronary arteries. To facilitate the understanding of the morphological details of the specimen, results are systematically presented following the nomenclature of the sixth edition of the *Nomina Anatomica Veterinaria* [30]. In accordance with conventional procedures and to facilitate the anatomic descriptions, verbs are used in the present tense.

The heart (*Cor*) is dorso-ventrally flattened; the *Facies auricularis* opposes the sternum and the *Facies atrialis* is continuous with the mediastinum. The *Basis cordis* is cranial and the two *Ventriculi cordis* are ventrocaudal. The external morphology shows the *Atria cordis* in one side (dorsal, *Facies atrialis*) (Figure 1), and their diverticulum or *Auricula atrii* in the opposite side (ventral, *Facies auricularis)* (Figure 2). The *Auricula dextra* partially covers the cranial aspect of the *Aorta ascendens* (Figure 2 and Figure 3), whereas the tip of the *Auricula sinistra* contacts the *Truncus pulmonalis* (Figure 2), which arises from the right side, from the *Ventriculus dexter*.

The two *Atria cordis* are separated by a common septum, the *Septum interatriale*, with a length of 13.25 cm and a width of 0.4 to 1.6 cm. In the right aspect of the septum, inside the *Atrium dextrum*, the CT scan showed that the *Fossa ovalis* forms a 4.30 cm^2^ (area) and 6.5 cm (perimeter) depression. This fossa is the remnant of the fetal *Foramen ovale* and surrounded by a prominent ridge, the *Limbus fossae ovalis*. The corresponding *Fossa ovalis* on the left aspect of the *Septum interatriale* is the *Valvula foraminis ovalis*, which is demonstrated in the CT images (Figure 5).

The limit between the *Basis cordis* and *Ventriculi cordis* is externally marked by the *Sulcus coronarius*, comprised of adipose tissue, which encircles the heart, except at the *Conus arteriosus*, where the *Truncus pulmonalis* arises (Figure 2 and Figure 3). No *os cordis* is present.

The limit (border) between *Facies auricularis* and *Facies atrialis* is clearly delineated over the ventricles by the *Margo ventricularis dexter*, part of the *Ventriculus dexter*, and the *Margo ventricularis sinister*, part of the *Ventriculus sinister*. Both borders converge ventrally in the *Apex cordis*, which incorporates the two ventricles. For this reason, there is no discernible *Incisura apices cordis* in the *Margo ventricularis dexter* (Figure 1 and Figure 2).

A longitudinal groove extends from the *Sulcus coronarius* and projects towards the apex on both sides of the heart. In the *Facies auricularis*, ventral to the *Conus arteriosus*, there is a *Sulcus interventricularis paraconalis* with a length of 30.13 cm (Figure 2), and on the *Facies atrialis*, the *Sulcus interventricularis subsinuosus* measures 22.47 cm (Figure 1).

These two grooves correspond to the outer projection of the *Septum interventriculare*. CT images of the septum identifies two portions, a smaller and thinner *Pars membranacea* located dorsally (length 4.72 cm and thickness 2.26 cm), and a main *Pars muscularis*, which internally separates both ventricles. The thickness of the *Pars muscularis* varies throughout its length; it is 9.56 cm (area 77.78 cm^2^) at level I, 4.72 cm (area 64.16 cm^2^) at level II, and 6.20 cm (area 49.07 cm^2^) at level III. In the sagittal plane, its length is 31.59 cm with an area of 184.45 cm^2^ (Figure 4).

The left and right ventricles are circumscribed by the parietal wall. In the right ventricle, the length of the parietal wall in the reference sagittal plane is 45.04 cm, the area is 126.79 cm^2^, and the ventricular luminal area is 325.59 cm^2^. In the reference transversal cross-sections, the corresponding length and area of the parietal wall and ventricle lumen cavity are as follows: Level I: 2.77 cm, 102.28 cm^2^, 197.18 cm^2^; Level II: 3.27 cm, 102.48 cm^2^, 145.99 cm^2^; Level III: 4.90 cm, 68.85 cm^2^, 36.17 cm^2^. Conversely, in the left ventricle, the reference sagittal plane wall length is 40.57 cm, the mural area is 187.87 cm^2^, and the cavity or luminal area is 133.05 cm^2^. In cross-sections, the length and area of the wall and ventricle lumen cavity are as follows: Level I: 5.06 cm, 184.04 cm^2^, 33.39 cm^2^; Level II: 6.29 cm, 132 cm^2^, 17.42 cm^2^; Level III: 5.28 cm, 124.42 cm^2^, 12.54 cm^2^ (Figure 4).

The CT and endoscopic images revealed intramyocardial vascular caverns within both the interventricular septum and the parietal walls of the ventricles (Figure 6, Figure 8A and Figure 9A). The estimated proportion of myocardial area occupied by these caverns in the interventricular septum is: 13.78% (I), 23.5% (II) and 7.49% (III); in the wall of the right ventricle: 69% (II) and 65.09%, (III); and for the wall of the left ventricle: 11.61%(I), 18.4% (II) and 17.89% (III).

Another unique feature of the killer whale heart is the numerous *Trabeculae carneae* found within the ventricles. Those trabeculae involve myocardium and endocardium and create a conspicuous and complex mesh of projections protruding into the lumina (Figure 8A).

The inner morphology of the *Atrium dextrum* and particularly the *Auricula dextra* also feature numerous trabeculae referred to as *Musculi pectinati*. Their thickness varies from a minimum of 0.23 cm to a maximum of 0.94 cm (Figure 7A). Since the external wall of the *Atrium dextrum* and the caudal vena cava were partly removed during the necropsy, relevant structures, such as the *Sinus coronarius*, *Sinus venarum cavarum* and *Sulcus terminalis* are not included in this specimen.

Between the *Atrium dextrum* and the *Ventriculus dexter*, the *Ostium atrioventriculare dextrum* has an internal diameter of 11.58 cm (CT sagittal plane). This orifice supports the *Valva atrioventricularis dextra (Valva tricuspidalis*) with three cusps, the *Cuspis angularis* (11.69 cm width), *Cuspis parietalis* (9.25 cm length) and *Cuspis septalis* (9.13 cm length) (Figure 3 and Figure 7B,C). The free edge of the cusps is attached to the three *Musculi papillares* of the right ventricle by fibromuscular cords (*Chordae tendineae)*. The common number of cords per cusp is 6–8 and the length of the chords range from 5.49 cm to 7.53 cm. Details of the *Musculi papillaris* were enhanced by endoscopy and CT. The *Musculus papillaris magnus* is located in the parietal wall with a thickness of 2.63 cm and a length of 8.20 cm, the *Musculi papillares parvi* is situated in the septum, and the *Musculus papillaris subarteriosus* is ventral to the *Ostium trunci pulmonali*.

Another relevant structure of the *Ventriculus dexter* is the *Trabecula septomarginalis*, which courses from *Septum interventriculare* to the parietal wall. This distinctive trabecula is not a unique fascicle, but consists of a principal fascicle with a length of 20.31 cm and a thickness of 4.70 cm with associated projections (Figure 8B).

The *Ventriculum dexter* communicates with the *Truncus pulmonaris* by the *Ostium trunci pulmonalis* (Figure 8C,D), which has an internal diameter of 8.64 cm. The lumen of this orifice is regulated by the *Valva trunci pulmonalis* which has three cusps, the *Valva semilunaris intermedia* (7.83 cm width), the *Valva semilunaris dextra* (4.93 cm length) and the *Valva semilunaris sinistra* (5.46 cm length). Despite a careful inspection with the endoscope, a *Noduli valvularum semilunarium* is not identified; however, *Lunulae valvularum semilunarium* are visible along the valve margins (Figure 8D).

The *Atrium sinistrum* (Figure 9A) is characterized by a conspicuous *Musculi pectinati* in the wall of the *Auricula sinistra*. The thickness of those muscles varies from a minimum of 0.32 cm to a maximum of 0.76 cm. During necropsy and extraction of the heart, the *Venae pulmonales* and part of the *Atrium sinistrum* were sectioned, and structures, such as the *Ostia venarum pulmonalium*, are not identified by endoscopy in CT scans.

The *Atrium sinistrum* communicates with the *Ventriculus sinister* through the *Ostium atrioventriculare sinistrum* (Figure 9B), which has a perimeter of 26.00 cm and an area of 6.20 cm^2^. In the ostium, there is a *Valva atrioventricularis sinistra* (*Valva bicuspidalis*, *mitralis*) with two cusps, the *Cuspis parietalis* and *Cuspis septalis*. The first cusp is attached to the parietal portion of the *Ostium atrioventriculare sinistrum* with a length of 7.85 cm and a width of 8.67 cm. The septal cusp also arises from the septal area of the *Ostium atrioventriculare sinistrum* with a length of 7.90 cm and width of 10.7 cm. (Figure 3 and Figure 9A,B) This last cusp establishes the lateral limit of the *Ostium aortae*, the opening of the left ventricle towards the *Aorta ascendens* (Figure 9D). The two mitral cusps have numerous *Chordae tendineae* attached to the apices of two *Musculli papillary*, both located in the parietal wall. The *Musculus papillaris subauricularis* has a length of 9.34 cm and a width of 2.72 cm, and the *Musculus papillaris subatrialis* has a length of 8.69 cm and a width of 3.64 cm (Figure 9B). As described in the *Ventriculum dexter*, a *Trabecula septomarginalis* is present within the *Ventriculus sinister*. This trabecula has a principal fascicle with a length of 4.07 cm length and width of 1.56 cm, that projects from the *Septum interventricularis* towads the *Musculli papillary*.

At the origin of the aorta, the inner diameter of the *Ostium aortae* is 8.55 cm and there is a *Valva aortae*, which is comprised of three cusps, the *Valvula semilunaris septalis*, *Valvula semilunaris dextra* and *Valvula semilunaris sinistra*. Due to silicone remnants from the coronary injection, measurements of these cusps as well as the *Sinus aortae*, and the origin of the *Arteriae coronaria dextra* and *sinistra* could not be determined. However, a description of the trajectory and measurements of the coronary vessels were possible because of the external dissection conducted during the curing of the specimen and radiopacity.

On dissection of the *Lamina visceralis* of the *Pericardum serosum*, the two coronary arteries have marked tortuosity and sinuosity. The *Arteria coronaria dextra* arises from above the *Valvula semilunaris dextra* with an approximate external diameter of 2.15 cm. Instead of a single vessel, this artery consists of several parallel vessels in the cranial and right trajectories of the *Sulcus coronarius* (Figure 1 and Figure 3). At the level of the *Sulcus interventricularis subsinuosus*, these vessels form the *Ramus interventricularis subsinuosus* (1.44 cm of external diameter), which diverges into several *Rami septales* that project deep into the *Septum interventriculare.*

The *Arteria coronaria sinistra* originates from above the *Valvula semilunaris sinistra* with an approximate external diameter at the origin of 2.93 cm that transitions to the *Ramus interventricularis paraconalis*, located in the *Sulcus interventricularis paraconalis* (Figure 2). This branch has an external diameter of 1.08 cm and also gives rise to some *Rami septales*, which project towards the *Septum interventriculae*. The *Arteria coronaria sinistra* also gives rise to another branch, the *Ramus circumflexus*, which progresses from the left and caudal trajectories of the *Sulcus coronarius* towards the right side of the heart. The diameter of this branch at its origin is 1.03 cm. As with the *Arteria coronaria dextra*, each part of the *Arteria coronaria sinistra* has several parallel vessels. No evidence of a *Ramus intermedius* for the *Margo ventricularis sinister* or a *Ramus interventricularis subsinosus* arising from the *Arteria coronaria sinistra* are observed.

## 4. Discussion

Due to the rarity of large cetacean strandings and post-mortem examination, there are few detailed investigations into the morphology of the heart [31,32], as reviewed in [33]. As with other cetaceans [34,35,36], the topography of the killer whale heart reflects the structure topography of the thorax, alignment of the ribs and their relatively unique bony distal part instead of the cartilage, which articulates with the sternum [12]. As noted by Slijper [14], the cetacean heart is rotated so that the left and right side of the heart are symmetric with the thorax and distinct to orientations of the heart within the thoracic cavity of terrestrial mammals. As a result, the heart is flattened dorso-ventrally with a ventral (sternal) aspect (*Facies auricularis*) that includes both ventricles, and a dorsal aspect (*Facies atrialis*) with the emergence of the aorta, the cranial and caudal vena cava and the pulmonary veins. The perspective of a flat (and not conic) heart is striking when the organ is excised and observed isolated from the lungs and the rib cage. In this sense, the shape and topography of the heart of delphinids is distinct to terrestrial artiodactyls, their closest evolutionary counterparts.

As previously reported, the *Foramen ovale* is nonpatent in adult cetaceans, albeit its closure occurs later postpartum in larger whales [9,37,38]. In our specimen, the *Fossa ovale* was well delineated, as in the bottlenose dolphin (see Figure 4.27 in [12]). Due to the dissection at necropsy, it was not possible to assess the persistence of the *Ductus arteriosus*; however, its loss has been shown in other adult cetacean species [38]. Similarly, the absence of the *Venae pulmonales* precluded assessment of the configuration of their opening into the left atrium, which differs in cetaceans from most terrestrial mammals [12,14]. The structure of the left atrium was typical of the heart of *Delphinidae* and other cetaceans [12,14]. The main characteristics were the evidence of an atrial portion, that it was rich in pectinate muscles and a very reduced venous component (*Sinus venosus*).

The *Musculi pectinati* in both the left and the right the atria form a dense network of muscle bundles that maintains the atrial walls opposed to one another [39]. This latter characteristic of the cetacean heart reduces the capacity of the atria, but presumably results in a stronger contraction, especially on the left side [14]. This apparent functional constraint may be compensated as each muscle unit spanning the atrial space includes an artery and a conduction bundle, which presumably participate in the rapid propagation of the contraction signal of the *Nodus sinuatrialis*. This anatomic adaptation is also noted in other large terrestrial mammals, including ungulates [39,40,41,42,43].

In the ventricles, the development of the *Trabeculae carnae* is also typically found in cetaceans and terrestrial artiodactyls. Their expansion is unique in the killer whale and other *Delphinidae*, such as the false killer whale [14] and the pilot whale [13]. This anatomical organization has been related to the synchronic velocity and smooth coordination of contraction [44], a very important feature in a large heart, correlated with lesser force and reduced contraction velocity [14]. The *Trabecula septomarginalis* (Tsm in Figure 7), reported in several cetaceans is believed to contain the bundle of His [14]. The sampling conditions did not allow us to determine the position and structure of the conducting system with precision. Data on dolphins and whales are scarce (for review, please refer to [12,45,46]) and available literature suggests that the innervation of the heart may differ between artiodactyls and humans [42].

Although some subepicardial plexi have been described in the porpoise [47], based on the extent of autolysis, the presence of large empty caverns in the myocardium of this killer whale heart may represent histolysis. It is difficult to confirm or infer a specific anatomical structure. During the freezing and fixation operations, the inner and outer surfaces were the first to freeze or in contact with the fixative; therefore, deep muscles within the myocardium may be the last to be affected by these processes. The diagnostic of mucormycosis implies fungemia, which could embolize with mycotic elements localized deep in more highly vascularized components of the myocardium.

As with other cetaceans, systematic evaluation of an archived plastinated killer whale heart revealed prominent coronary vasculature that is more prominent than documented in terrestrial mammals [13,31,39,48]. In contrast to coronary anastomoses described in larger cetacean species, such as the gray whale, the sei whale and the sperm whale [33], anastomoses of the coronary arteries in this killer whale heart and other delphinids, including harbor porpoises (*Phocoena phocoena*) [41] and bottlenose dolphins [39,49], have not yet been identified. A cause for the divergence of these vasculature patterns between larger and smaller cetaceans cannot be currently inferred. Additionally, although we are aware of the aortic bulb present in pinnipeds [50] and of the compliance of the aortic arch found in whales [51], we were not able to analyze the fresh aortic arch in the killer whale, since as seen in Figure 3, the aorta was cut close to the heart itself. Thickness measurements on the fixated heart were also deemed too uncertain to be made. This is a point which could be interesting to cover in further studies.

Although the data in this study involves a single heart, and therefore some peculiarities might not be the rule in all killer whales, the biological and medical value of these findings is important in that the heart of the killer whale follows what would be expected from a delphinid of its size. This implies that the reference points used in other species commonly kept such as *Tursiops truncatus* will likely be successfully transposed to killer whales. This holds true also for necropsy findings.

Due to the lack of available heart samples and associated physiologic and sonographic images [4], it is difficult to compare the findings from this individual animal to conspecifics and other species of the same family or suborder. Nevertheless, we believe the results from this study will be invaluable to clinicians, radiologists, anatomists and other research scientists interested in comparative or functional cardiac anatomy and physiology. It is imperative that comparative cardiac anatomy studies consider the implications of other systemic cardiovascular adaptations, such as of the caval sphincter in the diaphragm [52].

## 5. Conclusions

In conclusion, the anatomical description of the plastinated heart, supported by CT and endoscopic imaging, detailed specific adaptations of the killer whale heart to maintain the bursts of high energy activity of an apex predator. These anatomic features represent valuable baseline information to facilitate further biological, comparative and medical studies of the killer whale heart.

## Figures and Tables

**Figure 4 animals-12-00347-f004:**
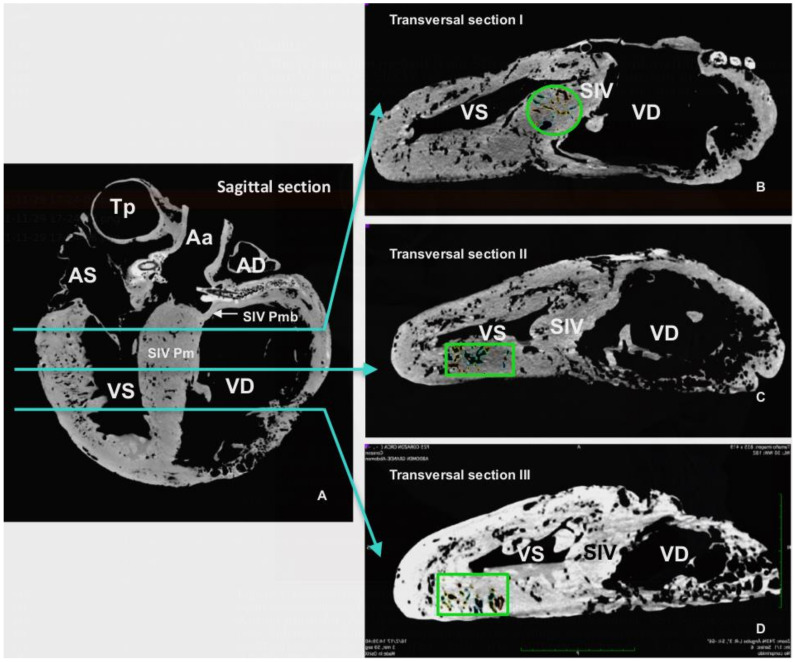
CT (computed tomography) sagittal and transversal sections used for morphometry. (**A**): Sagittal section through the *Sinus aortae*, where the two *Arteria carotis* arise. (**B**): Transversal section I, just distally to the *Valva atrioventricularis (dextra and sinistra).* (**C**): Transversal section II, 10 cm distally to previous section. (**D**): Transversal section III, 10 cm distally to section II. *Aorta ascendens* (Aa), *Truncus pulmonalis* (Tp), *Atrium dextrum* (AD), *Atrium sinistrum* (AS), *Ventriculus dexter* (VD), *Ventriculus sinister* (VS), *Septum interventriculare* (SIV): *Pars membranacea* (SIV Pmb) *Pars muscularis* (SIV Pm). The selected areas for estimatiion of cavern proportions showed in (**B**) (*Septum interventriculare*), and (**C**,**D**) (wall of *Ventriculus sinister*).

**Figure 5 animals-12-00347-f005:**
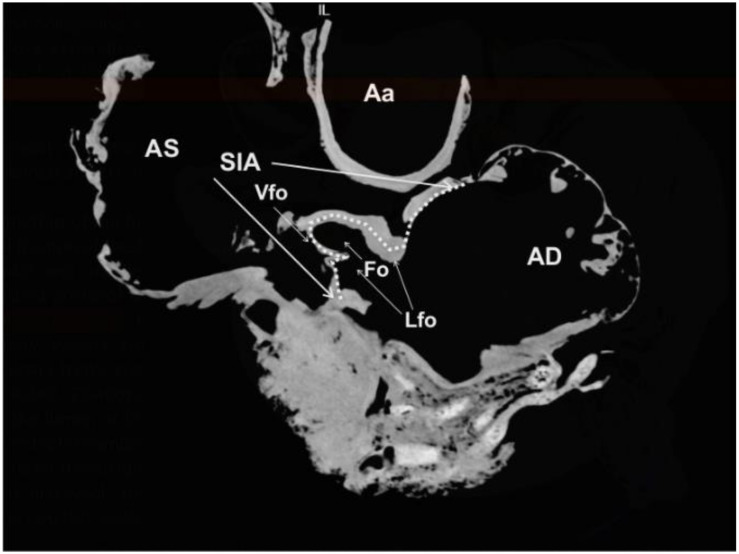
Plastinated killer whale heart. CT section at the level of the *Septum interatriale* (SIA, dotted line). *Aorta ascendens* (Aa), *Atrium dextrum* (AD), *Atrium sinistrum* (AS), *Fossa ovalis* (Fo), *Limbus fossae ovalis* (Lfo), *Valvula foraminis ovalis* (Vfo).

**Figure 6 animals-12-00347-f006:**
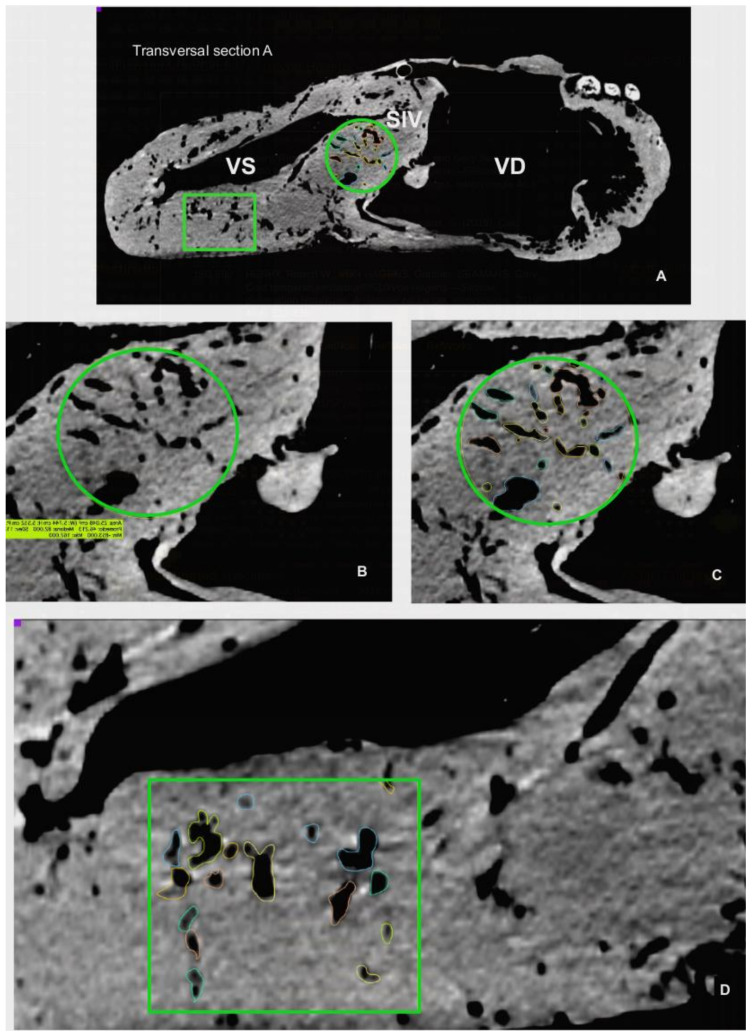
(**A**): Estimation of cavern proportion calculated at the CT transversal section I (Figure 4). (**B**): Magnification of the 25 cm^2^ green circled area showed in A. (**C**): Cavern estimation in selected round area of (**B**). (**D**): Cavern estimation in the 25 cm^2^ green squared area in the *Ventriculus sinister* (VS) wall showed in *Ventriculus dexter* (VD), *Septum interventriculare* (SIV).

**Figure 7 animals-12-00347-f007:**
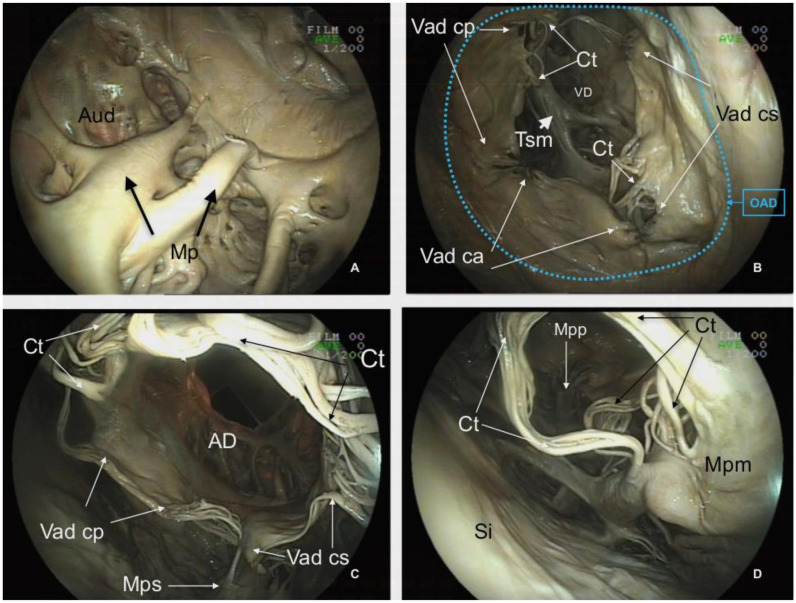
Endoscopic view of the *Atrium dextrum* (AD) and *Ventriculus dexter* (VD) of the plastinated killer whale heart. (**A**): View of the inner surface of the *Atrium dextrum*. (**B**): View of the *Valva atrioventricularis dextra* (Vad) from the *Atrium dextrum*. (**C**): View of the *Valva atrioventricularis dextra* from the *Ventriculus dexter*. (**D**): Details of the *Musculli papilaris*. *Auricula dextra* (Aud), *Musculi pectinati* (Mp), *Ostium atrioventriculare dextrum* (OAD), *Valva atrioventricularis dextra (Valva tricuspidalis): Cuspis angularis* (Vad ca), *Cuspis parietalis* (Vad cp), *Cuspis septalis* (Vad cs), *Chordae tendineae* (Ct), *Trabecula septomarginalis* (Tsm), *Musculus papillaris subarteriosus* (Mps), *Musculi papillares parvi* (Mpp), *Musculus papillaris magnus* (Mpm), *Septum interventriculare* (Si).

**Figure 8 animals-12-00347-f008:**
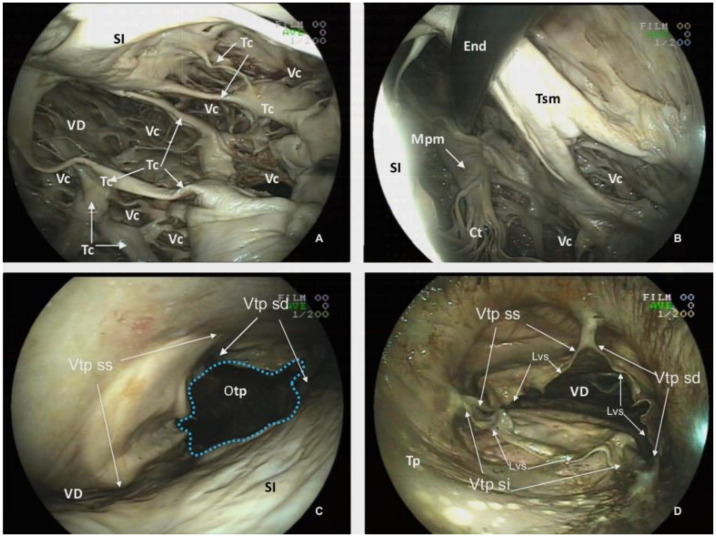
Endoscopic view of the *Ventriculus dexter* (VD) of the plastinated killer whale heart. (**A**) and (**B**): Lumen of the *Ventriculus dexter*. (**C**): Ventral view of the pulmonary truck valve from the left *Ventriculus sinister*. (**D**): Dorsal view of the *Valva trunci pulmonalis*, from the lumen of the *Truncus pulmonalis* (Tp). *Vascular caverns* (Vc), *Trabeculae carneae* (Tc) *Chordae tendineae* (Ct), *Trabecula septomarginalis* (Tsm), *Musculus papillaris magnus* (Mpm), *Septum interventriculare* (Si), *Ostium trunci pulmonalis* (Otp), *Valva trunci pulmonalis* (Vtp): *Valva semilunaris intermedia* (Vtp si), *Valva semilunaris dextra* (Vtp sd), *Valva semilunaris sinistra* (Vtp ss), Endoscope (End), *Lunulae valvularum semilunarium* (Lvs).

**Figure 9 animals-12-00347-f009:**
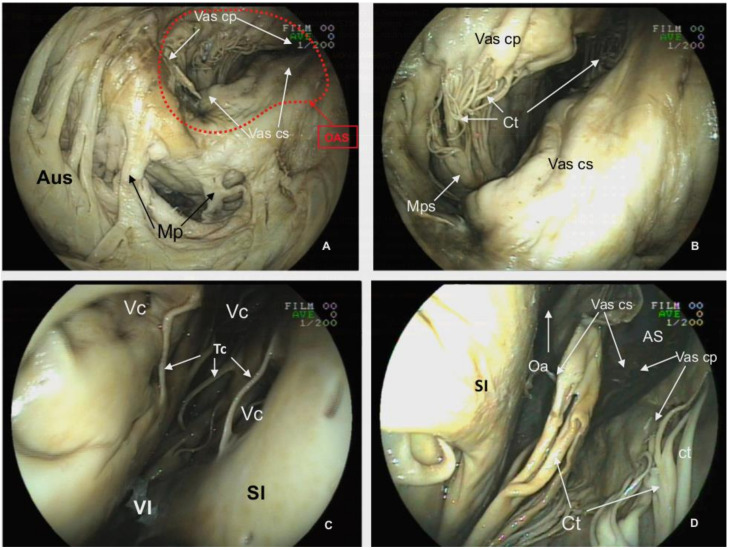
Endoscopic view of the *Atrium sinistrum* (AS) and *Ventriculus sinister* (VS) of the plastinated killer whale heart. (**A**): View of the inner surface of the *Auricula sinistra*. (**B**): View of the *Valva atrioventricularis sinistra* (Vas) from the *Atrium sinistrum*. (**C**): Lumen of the *Atrium sinistrum*. (**D**): Ventral view of the *Valva atrioventricularis sinistra* and the *Valva aortae*. *Auricula sinistra* (Aus), *Musculi pectinati* (Mp), *Ostium atrioventriculare sinistrum* (OAS), *Valva atrioventricularis sinistra* (Vas) (*Valva mitralis*): *Cuspis parietalis* (Vas cp), *Cuspis septalis* (Vas cs), *Musculus papillaris subauricularis* (Mps), *Chordae tendineae* (Ct), *Ventriculus sinister* (VI), Vascular caverns (Vc), *Trabeculae carneae* (Tc), *Septum interventriculare* (SI), *Ostium aortae* (Oa).

## Data Availability

Data can be made available upon reasonable request to authors.

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
