# Peer review of "The Heart of the Killer Whale: Description of a Plastinated Specimen and Review of the Available Literature"

_animals, 2022, doi:10.3390/ani12030347_

Round 1
Reviewer 1 Report
The data on this individual heart are new and the plastination and anatomy are new. This primary focus of the paper constitute a good contribution to the literature.
The killer whale heart measured here is from the Southern Resident population. The dive data in the table is from a Southern Hemisphere population. Anyway, compared to beaked whales of similar body size killer whales are not deep divers.
All the comparisons about blood volume and relative heart size among mammals are mostly wrong. These should be removed.
Line 72: I have read reference 1 and cannot find any justification for the statements about heart mass and blood volume. Heart mass of this killer whale relative to body mass is within the range of the bottlenose dolphin Tursiops truncatus [ref 13]. The general statements regarding blood volume and heart mass comparisons with terrestrial mammals are mostly wrong. There is considerable variability among mammals (see the review by Williams et al, 2015 cited below).
Geraci, J. R., Testaverde, S. A., & Aubin, D. S. (1978). A mass stranding of the Atlantic white-sided dolphin, Lagenorhynchus acutus: a study into pathobiology and life history. Marine Mammal Commission. (they measured 41 animals of both sexes and found a heart mass percentage of body mass of about 0.75 – higher than the one killer whale reported here.
Williams, T. M., Bengtson, P., Steller, D. L., Croll, D. A., & Davis, R. W. (2015). The healthy heart: lessons from nature's elite athletes. Physiology, 30(5), 349-357.
Author Response
Dear Editor,
We have received and read the comments made by the Reviewers, and we thank them for the time and effort they spend. We took care to implement the changes proposed and we trust that the manuscript is now significantly improved.
Please find below the comments of each reviewer and the responses we provided (in red).
Reviewer 1
The data on this individual heart are new and the plastination and anatomy are new. This primary focus of the paper constitute a good contribution to the literature.
The killer whale heart measured here is from the Southern Resident population. The dive data in the table is from a Southern Hemisphere population. Anyway, compared to beaked whales of similar body size killer whales are not deep divers.
Thank you for this comment. The difference between species and ecotypes is a fascinating one. There is undeniable differences in behavior and bodily features between ecotypes, although all the animals remain defined from the same species. While we do not pretend to define the whole species with one specimen (although in some cases it is all we have), we cannot assign these measurements and analysis to an ecotype only. Therefore, while the heart is that of a Southern Resident population, it will have to do for the species. There are very few anatomical differences reported between resident/coastal and pelagic Tursiops truncatus for example. Regarding “deep” diving, we do agree that the killer whale is not a regular diver to the 1000m mark, and the report of the Southern Hemisphere over 1000+m is more an illustration of what the species can do, with this type of heart, rather than a demonstration of recurrent diving behavior such as that found in beaked whales. We changed the text to clarify that point.
All the comparisons about blood volume and relative heart size among mammals are mostly wrong. These should be removed.
Line 72: I have read reference 1 and cannot find any justification for the statements about heart mass and blood volume. Heart mass of this killer whale relative to body mass is within the range of the bottlenose dolphin Tursiops truncatus [ref 13]. The general statements regarding blood volume and heart mass comparisons with terrestrial mammals are mostly wrong. There is considerable variability among mammals (see the review by Williams et al, 2015 cited below).
Geraci, J. R., Testaverde, S. A., & Aubin, D. S. (1978). A mass stranding of the Atlantic white-sided dolphin, Lagenorhynchus acutus: a study into pathobiology and life history. Marine Mammal Commission. (they measured 41 animals of both sexes and found a heart mass percentage of body mass of about 0.75 – higher than the one killer whale reported here.
Williams, T. M., Bengtson, P., Steller, D. L., Croll, D. A., & Davis, R. W. (2015). The healthy heart: lessons from nature's elite athletes. Physiology, 30(5), 349-357.
We thank the Reviewer for their suggestions. We are aware that there are several papers and discussions on the mammalian heart, possibly too many to be fitted into this short article dedicated to a never described before organ of a single species. The Review suggested in the revision process has been added. However the said interesting review draws mostly on larger baleen or sperm whales, and contains no data specifically referred to killer whales, To follow the spirit of the comment, we thus changed the paragraph, so that is now clear that we are not assigning specific data to the killer whale, but instead suggesting percentages, based on a wide range of possibilities found in the literature.
Reviewer 2 Report
This paper is well- written and describes an interesting multi-modal technique for the anatomical study of the killer whale heart. This information will help to fill a current data gap regarding cetacean heart anatomy. Below are my suggestions and edits:
Line 20, suggest rewording to say, “The four chambers, their content, respective openings, and communicating passages are defined based on a specimen…” as there are too many and’s in the current sentence.
Line 42 – suggest changing “such as transient killer whales predation of larger Mysticetes…” to “such as predation by transient killer whales of larger Mysticetes…” otherwise there should be an apostrophe after whales as it is currently written.
Table 1 is great!
Figure 1 caption –Italicize all of the Latin terms used.
All figure captions -- same as Fig 1, italicize the Latin anatomical terms (which are properly italicized in the body of the paper). Also, is there a reason the Latin versions of the anatomical terminology are being used in some places instead of the more commonly used clinical terms? For instance, why not use “right ventricle” instead of Ventriculus dexter? If the goal of the paper is to help clinicians, perhaps the more commonly used terms would be appropriate instead of the Latin terms? It would also be good to be more consistent throughout the paper one way or the other. In some of the paragraphs, and the endoscopic figure descriptions the commonly used terms are utilized, but in the gross figure captions and CT figures the Latin anatomical names for the same thing are used. If this is intentional, please elaborate on it. In my opinion, this adds unnecessary confusion to use the Latin terms in some instances and the common terms in others, but if this is being done intentionally please elaborate on it. Otherwise, would try to be more consistent throughout the paper.
Line 445 – should be “His” not “Hiss”
Line 449 – can utilize more references relating to cetacean electrical conductivity, which describe cetacean hearts as Class B ventricular activation: Harms et al 2013 JZWM “ELECTROCARDIOGRAMS OF BOTTLENOSE DOLPHINS (TURSIOPS TRUNCATUS) OUT OF WATER: HABITUATED COLLECTION VERSUS WILD POSTCAPTURE ANIMALS” and Detweiler, D. K. 1989. The mammalian electro- cardiogram: comparative features. In: MacFarlane, P. W., and T. D. Veitch Lawrie (eds.). Comprehensive Electrocardiology: Theory and Practice in Health and Disease. Pergamon Press, New York, New York. Pp. 1331–1377.
Line 469 – sentence does not make sense as written. Change from “single heart, and the biological…” to a single heart, the biological”
Line 485 – consider re-wording the conclusion… the word “resistant “ as it does not seem to be the appropriate word here, and the conclusion as written is not summarizing the rest of the paper well. le. table 1 is not the most important take-home message of this paper, and it would be better to emphasize the take-home message of this paper and what new information has been presented.
Consider rewording the conclusion to something along the lines of “ In conclusion, the anatomic description of the plastinated heart described herein, utilizing CT and endoscopic studies, details specific traits and adaptations of the killer whale heart to maintain the bursts of high energy demands of a formidable marine predator. These results may provide valuable baseline information to facilitate further biological and medical studies of the killer whale heart.
Author Response
Dear Editor,
We have received and read the comments made by the Reviewers, and we thank them for the time and effort they spend. We took care to implement the changes proposed and we trust that the manuscript is now significantly improved.
Please find below the comments of each reviewer and the responses we provided (in red).
Reviewer 2
This paper is well-written and describes an interesting multi-modal technique for the anatomical study of the killer whale heart. This information will help to fill a current data gap regarding cetacean heart anatomy. Below are my suggestions and edits:
Line 20, suggest rewording to say, “The four chambers, their content, respective openings, and communicating passages are defined based on a specimen…” as there are too many and’s in the current sentence.
Thank you, this oversight has been corrected.
Line 42 – suggest changing “such as transient killer whales predation of larger Mysticetes…” to “such as predation by transient killer whales of larger Mysticetes…” otherwise there should be an apostrophe after whales as it is currently written.
The sentence has been changed.
Table 1 is great!
Figure 1 caption –Italicize all of the Latin terms used.
We agree with the reviewed. We believe there might have been a mismatch in the final version sent to the reviewers. In the version asked by the Academic Editor, the latin names had indeed been italicized. In any case, the present revised version corresponds to the Reviewer’s request.
All figure captions -- same as Fig 1, italicize the Latin anatomical terms (which are properly italicized in the body of the paper). Also, is there a reason the Latin versions of the anatomical terminology are being used in some places instead of the more commonly used clinical terms? For instance, why not use “right ventricle” instead of Ventriculus dexter? If the goal of the paper is to help clinicians, perhaps the more commonly used terms would be appropriate instead of the Latin terms? It would also be good to be more consistent throughout the paper one way or the other. In some of the paragraphs, and the endoscopic figure descriptions the commonly used terms are utilized, but in the gross figure captions and CT figures the Latin anatomical names for the same thing are used. If this is intentional, please elaborate on it. In my opinion, this adds unnecessary confusion to use the Latin terms in some instances and the common terms in others, but if this is being done intentionally please elaborate on it. Otherwise, would try to be more consistent throughout the paper.
As requested by the Academic Editor of the Journal shortly after submission, a second version of the manuscript with some recommendations had to be prepared. In such second version inconsistencies in the use of anatomical nomenclature were also detected by the authors and corrected (most probably, the reviewer was provided with the first submitted version of the paper).
Authors agreed that the best option was to use the Nomina Anatomica Veterinaria, so latin names in italics were exensively used in the second version of the manuscript. Being such, the paper will contribute to the scientific community (clinicians included) with a sound and accurate anatomical description of this unique specimen.
Line 445 – should be “His” not “Hiss”
This was corrected, thank you
Line 449 – can utilize more references relating to cetacean electrical conductivity, which describe cetacean hearts as Class B ventricular activation: Harms et al 2013 JZWM “ELECTROCARDIOGRAMS OF BOTTLENOSE DOLPHINS (TURSIOPS TRUNCATUS) OUT OF WATER: HABITUATED COLLECTION VERSUS WILD POSTCAPTURE ANIMALS” and Detweiler, D. K. 1989. The mammalian electro- cardiogram: comparative features. In: MacFarlane, P. W., and T. D. Veitch Lawrie (eds.). Comprehensive Electrocardiology: Theory and Practice in Health and Disease. Pergamon Press, New York, New York. Pp. 1331–1377.
The references were added, thank you.
Line 469 – sentence does not make sense as written. Change from “single heart, and the biological…” to a single heart, the biological”
Thank you, the sentence was corrected.
Line 485 – consider re-wording the conclusion… the word “resistant “ as it does not seem to be the appropriate word here, and the conclusion as written is not summarizing the rest of the paper well. le. table 1 is not the most important take-home message of this paper, and it would be better to emphasize the take-home message of this paper and what new information has been presented.
The conclusion was started over. We summarized our own results and concluded.
Consider rewording the conclusion to something along the lines of “ In conclusion, the anatomic description of the plastinated heart described herein, utilizing CT and endoscopic studies, details specific traits and adaptations of the killer whale heart to maintain the bursts of high energy demands of a formidable marine predator. These results may provide valuable baseline information to facilitate further biological and medical studies of the killer whale heart.
Thank you very much for this suggestion. We rewrote the conclusion based on your text.
Reviewer 3 Report
Overall, the paper was well written and scientifically sound. It was an interesting read. However, there were several places in the paper that need improvement for this paper to be published.
Line 41-43. A long confusing sentence. Needs to be reworked.
Line 44. “Deep diving is not fundamental for orcas,…” What is meant by fundamental? Awkward and confusing
Lines 48-51. This seems to be just placed. It doesn’t offer much and I was confused at the value this paragraph. Either expand or delete.
Table 1. have ocean dive vs ocean dives. Be consistent
Line 58- need citation(s) at the end of the sentence
Line 88-89. “… performed by a standard killer..” the a makes it seem like there are different necropsy protocols.
Line 397-398. This isn’t shocking news that the heart of a mammal is like other mammals. This sentence should be deleted.
Line 417. The authors abruptly ended this paragraph. Considering the paragraph is based on a reference from the 1930s, is this paragraph really offering substantial intellectual value? I did not get anything out of it except confusion.
Line 469-472. Well, what is the value of this paragraph related to the results? It just seems like I am reading writing for the sake of the writing. It does not explain why “the medical value of these findings is important” (Line 470). How are the results from this paper applicable to vets in marine theme parks around the world? Tell us what you want us to know.
Line 474- Glad this was mentioned.
Line 482-486 (Conclusion). This was a terrible way to end the paper. This had me going back to the table that was not your own data. This needs to be completely redone. I would suggest ending on the importance of the study and a summary of the results. Not angering the reader by having them go back to look at data not collected in this study to see what the heart does. Redo this section or just delete it entirely. It needs to go.
Author Response
Dear Editor,
We have received and read the comments made by the Reviewers, and we thank them for the time and effort they spend. We took care to implement the changes proposed and we trust that the manuscript is now significantly improved.
Please find below the comments of each reviewer and the responses we provided (in red).
Reviewer 3
Line 41-43. A long confusing sentence. Needs to be reworked.
This sentence was also mentioned by Reviewer 2. We edited the sentence accordingly.
Line 44. “Deep diving is not fundamental for orcas,…” What is meant by fundamental? Awkward and confusing
Thank you, the sentence has been edited to acknowledge the fact that most killer whales do not need to dive as deep as this recorded dive; however some have this capacity.
Lines 48-51. This seems to be just placed. It doesn’t offer much and I was confused at the value this paragraph. Either expand or delete.
We reduced this to a simple sentence mentioning the relative lack of anatomical studies on killer whales.
Table 1. have ocean dive vs ocean dives. Be consistent
thank you, this typo was corrected.
Line 58- need citation(s) at the end of the sentence
This has been mended.
Line 88-89. “… performed by a standard killer..” the a makes it seem like there are different necropsy protocols.
This was edited out.
Line 397-398. This isn’t shocking news that the heart of a mammal is like other mammals. This sentence should be deleted.
This sentence was removed.
Line 417. The authors abruptly ended this paragraph. Considering the paragraph is based on a reference from the 1930s, is this paragraph really offering substantial intellectual value? I did not get anything out of it except confusion.
The paragraph was deleted. The intention was to give some of the historical perspective on hypotheses mentioned in the past.
Line 469-472. Well, what is the value of this paragraph related to the results? It just seems like I am reading writing for the sake of the writing. It does not explain why “the medical value of these findings is important” (Line 470). How are the results from this paper applicable to vets in marine theme parks around the world? Tell us what you want us to know.
The clinical and biological relevance of these findings was explained further.
Line 474- Glad this was mentioned.
Line 482-486 (Conclusion). This was a terrible way to end the paper. This had me going back to the table that was not your own data. This needs to be completely redone. I would suggest ending on the importance of the study and a summary of the results. Not angering the reader by having them go back to look at data not collected in this study to see what the heart does. Redo this section or just delete it entirely. It needs to go.
The conclusion was started over. We summarized our own results and concluded.
Round 2
Reviewer 1 Report
Lines 67 to 74. The paragraph is about Delphinidae, yet they put in a reference 46 that is about the huge fin whale to support a statement that “heart mass and blood volume are lower than in terrestrial mammals. With respect to Delphinidae, this is wrong. McAlpine found that sperm whales had smaller hearts that terrestrial mammals relative to body size but that faster moving fin whales had hear sizes similar to terrestrial mammals relative to body size.
Line 67: “The total mass of circulating blood of members of the Family Delphinidae is approx. 7.4% of total body mass (thus between 20 to 60 liters in the killer whale) [13; 14].” This generalization is incorrect. Ref 13 found blood volume in Tursiops truncatus to be 6.5 to 8.3% whereas in a faster swimmer L obliquidens it ranged from 9.5 to 11.8%.
Lines 68-70: The weight of the normal heart in the bottlenose dolphin is 0.93% of total body weight in the 69 bottlenose dolphin and approx. 0.5 to 1.3% of the total body mass in other adult members 70 of the Family (thus between 15 and 40 kg in the killer whale) [8, 10, 13, 47].” Wrong, Ref 13 found heart mass of 0.50 to 0.56 % of body mass. Whereas the same reference found L. obliquidens to have heart mass of 0.70 to 0.92 % of body mass. Williams et al gave an average figure for all dolphins of about 0.93%.
Line 71 and 72: “Both the heart mass and blood volume have been reported to be lower than those of terrestrial mammals 72 [46].” Wrong, Only the sperm whale had a lower heart mass that terrestrials. Blood volumes in many cetaceans are higher than terrestrial mammals. Sleet et al found very high blood volumes in the sperm whale. Blood volume and heart mass are related in some small delphinids and phocoenids but not in many larger species.
Line 73: “However, more recent work seems to show a certain variability [1, 45; for review on the mammalian heart see also 45]. “ Wrong Ref 13 is from the 1960s and a number of other measures are not recent.
The authors have no measure of body mass of this killer whale. To their credit they measure heart mass. This one measure does not justify the incomplete generalizations of lines 67 to 75. These generalizations are not supported by the data. The paper would be improved greatly by deletion of this paragraph.
Author Response
Dear Reviewer,
after reading your second review, we agreed that simply deleting the paragraph would solve your doubts and keep the focus on the data we gathered.
Thank you for your constructive criticism.
Sincerely,
Jean-Marie Graic, corresponding author